# Computational-Based Discovery of the Anti-Cancer Activities of Pyrrole-Based Compounds Targeting the Colchicine-Binding Site of Tubulin

**DOI:** 10.3390/molecules27092873

**Published:** 2022-04-30

**Authors:** Sergei Boichuk, Kirill Syuzov, Firuza Bikinieva, Aigul Galembikova, Svetlana Zykova, Ksenia Gankova, Sergei Igidov, Nazim Igidov

**Affiliations:** 1Department of Pathology, Kazan State Medical University, 420012 Kazan, Russia; grop2019@gmail.com (K.S.); f.bikinieva@kazangmu.ru (F.B.); ailuk000@mail.ru (A.G.); 2Department of Radiotherapy and Radiology, Russian Medical Academy of Continuous Professional Education, 125993 Moscow, Russia; 3Biologically Active Terpenoids Laboratory, Kazan Federal University, 18 Kremlyovskaya St., 420008 Kazan, Russia; 4Department of Pharmacology, Perm State Academy of Pharmacy, 614990 Perm, Russia; zykova.sv@rambler.ru (S.Z.); gankova.ksenia@mail.ru (K.G.); s.igidov@mail.ru (S.I.); igidov_nazim@mail.ru (N.I.)

**Keywords:** microtubules, ethyl-2-amino pyrrole-based carboxylates (EAPCs), tubulin depolymerization, cell cycle, mitotic arrest, apoptosis, breast, lung, cancer, paclitaxel, vinblastine, colchicine, induced-fit docking, binding metadynamics, unbiased molecular dynamics

## Abstract

Despite the tubulin-binding agents (TBAs) that are widely used in the clinic for cancer therapy, tumor resistance to TBAs (both inherited and acquired) significantly impairs their effectiveness, thereby decreasing overall survival (OS) and progression-free survival (PFS) rates, especially for the patients with metastatic, recurrent, and unresectable forms of the disease. Therefore, the development of novel effective drugs interfering with the microtubules’ dynamic state remains a big challenge in current oncology. We report here about the novel ethyl 2-amino-1-(furan-2-carboxamido)-5-(2-aryl/tert-butyl-2-oxoethylidene)-4-oxo-4,5-dihydro-1H-pyrrole-3-carboxylates (EAPCs) exhibiting potent anti-cancer activities against the breast and lung cancer cell lines in vitro. This was due to their ability to inhibit tubulin polymerization and induce cell cycle arrest in M-phase. As an outcome, the EAPC-treated cancer cells exhibited a significant increase in apoptosis, which was evidenced by the expression of cleaved forms of PARP, caspase-3, and increased numbers of Annexin-V-positive cells. By using the in silico molecular modeling methods (e.g., induced-fit docking, binding metadynamics, and unbiased molecular dynamics), we found that EAPC-67 and -70 preferentially bind to the colchicine-binding site of tubulin. Lastly, we have shown that the EAPCs indicated above and colchicine utilizes a similar molecular mechanism to inhibit tubulin polymerization via targeting the T7 loop in the β-chain of tubulin, thereby preventing the conformational changes in the tubulin dimers required for their polymerization. Collectively, we identified the novel and potent TBAs that bind to the colchicine-binding site and disrupt the microtubule network. As a result of these events, the compounds induced a robust cell cycle arrest in M-phase and exhibited potent pro-apoptotic activities against the epithelial cancer cell lines in vitro.

## 1. Introduction

The microtubules are known as important regulators of many aspects of cellular functions, including cell proliferation and migration, vesicle trafficking during endocytosis, chromosomal segregation during mitosis, etc. [1,2,3,4]. The last one has made them an attractive target for the development of anti-cancer drugs. The protein tubulin, a major compartment of the microtubules, contains several various binding sites for the small-molecule drugs. This includes the laulimalide, maytansine, taxane/epothilone, vinca alkaloid, and colchicine sites [5,6,7,8,9,10]. Based on their effects on the microtubule dynamic state, the tubulin-binding agents (TBAs) are categorized into 2 major groups: microtubule-stabilizing agents including the taxanes, epothilones, and laulimalide and microtubule-destabilizing agents that are composed of colchicine, the vinca alkaloids, and maytansine [11,12].

Despite the impressive response rate shortly after the initiation of taxanes- and/or vinca alkaloids-based therapies, the extended use of these chemotherapeutic agents is limited due to the acquired resistance of tumors to these drugs. This might be due to the broad spectrum of molecular mechanisms: e.g., overexpression of ABC-transporters, such as P-glycoprotein in tumor cells [13,14], mutations in β-tubulin, and the altered expression of specific β-tubulin isotypes [15,16], etc. In this view, the agents targeting the colchicine-binding site (CBS) exhibit several advantages, including overcoming multidrug resistance (MDR), which is supplemented by their well-known abilities to inhibit angiogenesis [17,18,19,20]. Moreover, CBS inhibitors (CBSi) have several significant advantages over other microtubule-targeting agents (MTAs) because this site of tubulin is more amenable to the molecules with favorable physicochemical properties that improve oral bioavailability over taxanes and vinca alkaloids binding sites, for example, have less drug-drug interactions, and are less prone to developing multi-drug resistance. Despite the clinical use of colchicine as an anti-cancer derivative, it is currently excluded based on its extreme toxicity; the broad number of CBSi is currently described [21] and can be used as structurally diverse scaffolds for the generation of potent anti-cancer derivatives exhibiting limited toxicities, improved solubility, etc. [22].

In general, the scaffolds to develop the novel TBAs targeting the CBS are based on the presence of diverse chemical groups, including the trimethoxyphenyl (TMP) group. This principle is raised from the structure of colchicine, which consists of a 3,4,5-trimethoxyphenyl ring (the A ring), a saturated seven-membered ring containing an acetamido group at position 7 (the B ring), and a tropolone ring (the C ring). It was shown that the TMP group of colchicine is oriented within β-tubulin close to Cys β241 and thereby allows colchicine to bind to β-tubulin at its interface with α-tubulin, subsequently inhibiting tubulin polymerization [23]. Thus, it was accepted that the most common conservative chemical groups in the compounds exhibiting tubulin depolymerizing properties are composed of trimethoxyphenyl (TMP) moiety (ring A) and the linker region connecting ring A and ring B [24,25,26]. This point of view was mainly based on the structure-activity relationship (SAR) analysis of colchicine and combretastatin A-4 and its analogs, the well-known earliest inhibitors of tubulin polymerization.

The present study aimed to examine the cytotoxic activities of the 2-amino-1-(furan-2-carboxamido)-5-(2-aryl/tert-butyl-2-oxoethylidene)-4-oxo-4,5-dihydro-1H-pyrrole-3-carboxylates (EAPCs) synthesized in our lab and further assess their molecular mechanisms of anti-cancer activities as the novel chemical structures exhibiting tubulin-binding activities.

Here, we show for the first time that 2-aminopyrrole derivatives (EAPC-67 and -70) exhibit potent cytotoxic activities against a broad spectrum of cancer cell lines (e.g., triple-negative breast cancer HCC 1806, MDA-MB-231; non-small cell lung cancer H 1299, etc.). This was evidenced by the decreased viability of EAPC-treated cancer cells as measured by MTS-based assay, increased expression of apoptotic markers (e.g., cleaved forms of PARP, caspase-3), and the number of apoptotic (e.g., Annexin V-positive) cells. These activities of EAPCs were due to their abilities to halt a cell cycle progression and induce a robust cell cycle arrest in M-phase. This, in turn, was a consequence of the inhibition of the microtubule polymerization due to the ability of EAPCs to bind with the CBS of the tubulin. To reveal this possibility, we used different in silico molecular modeling methods. First, we used a blind docking strategy to determine the preferred drug-binding site on the tubulin dimer. We found the CBS of tubulin to be the most energetically favorable for EAPCs indicated above. Next, we used the induced docking of the ligands in the CBS. By utilizing this approach, we identified two of the most probable poses of EAPC-67 exhibiting the maximal induced-fit docking (IFD) scores. Afterward, by using the binding metadynamics approach, we determined the stability of the poses for these selected ligands by calculating the PoseScore and CompScore indexes. Finally, based on the metadynamics data, we performed 100 nanosecond unbiased molecular dynamics to determine the key interactions of the most stable ligand pose with the CBS. Data analysis of the induced fit docking and the trajectory of molecular dynamics revealed that the external orientation of the T7 loop of the β-chain of tubulin is required for the successful binding of the drug to the CBS. Otherwise, steric clashes are created between the ligand atoms and the amino acid atoms of the T7 loop that block access for the ligand to its binding site. This data indicates that the molecular mechanism of action of EAPCs might be similar to colchicine and is predominantly associated with the impossibility of internal turning of the T7 loop in the β-chain of tubulin. Based on our data, we also proposed that EAPCs block the rotation of the T7 loop and subsequently prevent the conformational changes in the tubulin dimer from the curved to the straight state which is required for tubulin polymerization.

## 2. Materials and Methods

### 2.1. Chemical Compounds

Paclitaxel (PTX), vinblastine (Vin), and colchicine (Col) were purchased from Sigma, St Louis, Missouri, USA, and dissolved in 100% dimethyl sulfoxide (DMSO) (Sigma Aldrich, St. Louis, MO, USA). 10 novel 2-amino-1-(furan-2-carboxamido)-5-(2-aryl/tert-butyl-2-oxoethylidene)-4-oxo-4,5-dihydro-1H-pyrrole-3-carboxylates (EAPCs) as potential microtubule destabilizing agents were synthesized in our laboratory according to the standard protocols, as shown in Figure 1 and in previous studies [27].

### 2.2. Molecular Modeling Methods

Blind docking: To identify the potential binding sites of EAPCs on the tubulin, the molecular docking procedure was performed by using Schrodinger molecular modeling software (Schrödinger Inc., New York, NY, USA, 2021). The Glide Docking XP mode was used to find the tubulin-ligand supramolecular complex with the minimal energy scoring function. The tubulin structure (no. 4O2B) was downloaded from the Protein Data Bank (PDB) (4O2B. Available online: www.rcsb.org/structure/4O2B, accessed on 3 April 2022). The protein structure was prepared before docking using the Protein Preparation Wizard module. Ligand preparation was performed by the standard protocol using the module LigPrep [28]. To determine the possible interactions of the most active compounds with the tubulin molecule, “blind docking” was performed. Only tubulin inhibitor binding sites were used for analysis. After defining binding sites, a Grid Box for each of them was generated by using the receptor grid module.

Induced Fit Docking: The Standard Induced Fit Docking Protocol is used to generate possible induced CBS conformations [29]. Initially, the prepared ligands were docked using a softened potential with Glide XP [30]. The Coulomb-vdW scaling factor was set to 0.5 for the ligand and protein atoms, and a maximum of 20 poses were generated on one ligand tautomer. The generated poses were further processed by Prime for side-chain refinements within 7 Å of the binding site for better accommodation of the ligands. Next, the protein-ligand system was minimized using the OPLS4 force field [31]. In the final stage, the ligands were docked using the Glide XP module into the optimized binding site generated within 30 kcal mol^−1^ of the lowest energy structure obtained after Prime refinement to generate 20 poses per system. To rank the poses of the ligands, we used IFD score (IFDScore = 1.0 × GlideScore + 0.05 × PrimeEnergy). Based on the maximum IFD score the conformations of the ligand-protein complex were chosen for the next steps of molecular modeling.

Metadynamics: The metadynamics method was used to assess the stability of the EAPC-67 pose in the selected induced-fit conformations. 10 independent metadynamics simulations of 10 ns are performed using root-mean-square deviation (RMSD) of the ligand heavy atoms as the collective variable. The alignment before the RMSD calculation was done by selecting protein residues within 3 Å of the ligand. The hill height and width were set to 0.05 kcal/mol (about 1/10 of the characteristic thermal energy of the system, *kBT*) and 0.02 Å, respectively [32]. To remove bad contacts and/or strain in the initial starting structure and to slowly reach a temperature of 300 K for the simulated system, a 0.5 ns simulation was performed in the NVT ensemble before the start of the metadynamics itself. The EAPC-67-tubulin complex was solvated in a simple point charge (SPC) water model using an orthorhombic box with periodic boundary conditions. The total charge of the system was neutralized by the addition of sodium and potassium ions. The final snapshot of the short unbiased molecular dynamics (MD) simulation of 0.5 ns is then used as the reference structure for following the metadynamics production.

Molecular dynamics: After the metadynamics procedure, 100 ns unbiased molecular dynamics was performed to assess the position of the ligand in the CBS and determine key amino acid interactions over time. The Desmond program in Schrödinger suite 2021-2 was used to perform classical molecular dynamics of the EAPC-67-tubulin complex [33]. The EAPC-67-tubulin complex was solvated in a simple point charge (SPC) water model using an orthorhombic box with periodic boundary conditions. The total charge of the system was neutralized by the addition of sodium and potassium ions. To minimize and relax the system, a short simulation in the NVT ensemble of 100 picoseconds was performed. The main simulation was performed in the NPT ensemble. The simulation was run for a total of 100 ns with a recording interval of 100 ps. The temperature and pressure were kept constant at 300 K and 1.01325 bar, respectively, throughout the simulations. Calculation of the root means square deviations (RMSD) and analysis of the protein-ligand interaction were performed using the simulation interaction diagram (SID) program tool in the Schrödinger Maestro Suite.

Trajectory analysis and clusterization: To investigate conformational flexibility over MD simulation, the collective motions of the EAPC-67-tubulin complex were investigated for all snapshots in the trajectories. For the extraction of snapshots from 100 nanosecond molecular dynamics, VMD software was used [34]. After snapshot extraction using the *pca.xyz()* function in the Bio3D package in R, we performed a principal component analysis (PCA) analysis over cartesian coordinates of C*α* atoms of all snapshots in the trajectories [35,36]. Each trajectory was projected onto the first eigenvector subspaces to reveal concerted atomic displacements, representing the “essential dynamics” of the protein. Using the PCA data, K-means clusterization analysis was performed over each trajectory described by the first three principal components. The optimal number of clusters was calculated using the Silhouette method [37]. After clustering, a medoid was selected from each cluster as a representative conformation.

The molecular mechanics/generalized Born surface area (MM-GBSA): The MM-GBSA method was used as a post-molecular dynamic validation tool [38]. Based on the medoids of clusters, we calculated the binding free energy (∆Gbind) of each EAPC-67-tubulin complex, including the solvation energy (∆Gsolv), using the implicit water model. A representative conformation of the ligand-receptor complex was chosen based on minimal ΔGbind Prime energy.

### 2.3. Cell Lines and Culture Conditions

The human basal-like TNBC cell lines HCC 1806, MDA-MB-231, and non-small cell lung cancer cell line H1299 were purchased from the American Type Culture Collection (Manassas, VA, USA). The cell lines indicated above were maintained in RPMI-1640 medium (Paneco, Moscow, Russia) supplemented with 15% fetal bovine serum (Gibco; Thermo Fisher Scientific Inc., Waltham, MA, USA), 50 U/mL penicillin, and 50 μg/mL streptomycin. The cell lines were cultured at 37 °C in a humidified atmosphere of 5% CO_2_ in an incubator (LamSystems, Miass, Russia).

### 2.4. Antibodies

The primary antibodies used for western blotting were anti-PARP (cat. no. 436400; Invitrogen; Thermo Fisher Scientific Inc., Waltham, MA, USA), cleaved caspase-3 (cat. no. 9661S), phospho-NuMA Ser395 (cat. no. 3429), phospho-Histone H3 Ser10 (cat. no. 53348) (Cell Signaling Technology Inc., Danvers, MA, USA), and beta-actin (cat. No. A00730-200, GenScript, Piscataway, NJ, USA); HRP-conjugated secondary antibodies, anti-mouse immunoglobulin (Ig)G (cat. no. sc-2005) and anti-rabbit IgG (cat. no. sc-2004), were purchased from Santa Cruz Biotechnology, Dallas, TX, USA.

### 2.5. Western Blotting Analysis

To examine the protein expression in parental and Tx-R cells, whole-cell lysates (WCL) were prepared by scraping the cells growing as monolayers into a radio-immunoprecipitation buffer (RIPA buffer) (25 mM Tris-HCl pH 7.6, 5 mM EDTA, 150 mM NaCl, 0.1% SDS, 1% NP-40, 1% sodium deoxycholate) supplemented with the cocktail of protease and phosphatase inhibitors. The cellular lysates were further incubated for 1 h at 4 °C and clarified by centrifugation for 30 min at 11,400 rpm at 2 °C. The protein concentrations in WCL were calculated by the Bradford assay. The protein samples (20 µg) were loaded on the 4–12% Bis-Tris or 3–8% Tris-acetate NuPAGE gels (Invitrogen, Carlsbad, CA, USA) and upon completion of electrophoresis transferred to a nitrocellulose membrane (Bio-Rad, Hercules, CA, USA), membranes were probed with primary (1:1000 and incubated overnight at 4 °C), and secondary antibodies (1:1000 and incubated for 1 h at room temperature) and visualized by enhanced chemiluminescence (Western Lightning Plus-ECL reagent, Perkin Elmer, Waltham, MA, USA).

### 2.6. Tubulin Polymerization Assay

The impact of the CAs on the dynamic state of the tubulin polymerization was assessed by using the Tubulin Polymerization kit (Cytoskeleton Inc., Denver, CO, USA) as specified by the manufacturer. Results were obtained on a SpectraFluor Plus microplate reader (Tecan GmbH, Grödig, Salzburg, Austria) and readings were taken every minute for 1 h (61 measurements in total).

### 2.7. Flow Cytometry

The number of apoptotic cells was counted by using Muse Annexin V Dead Cell Kit (Merck KGaA, Darmstadt, Germany) according to the manufacturer’s instructions. Unstained and single-stained untreated cells were used as controls. Cells were analyzed in a Muse Cell Analyzer (Merck KGaA, Darmstadt, Germany). For all the experiments indicated above at least 10,000 events were acquired for each sample. Results were presented as the percentage of desired cells relative to the total number of cells as mean ± standard deviation of four biological repeats.

### 2.8. Statistics

All the experiments have been repeated a minimum of 3 times. The results are presented as the mean ± standard error (SE) for each group. Statistical analyses (Student’s *t*-test) were performed using Statistical software program version 7.0 (S.A. Glantz, McGraw Hill Education, New York, NY, USA). *p* < 0.05 was considered to indicate a statistically significant difference.

## 3. Results

### 3.1. Synthesis of 2-Aminopyrrole Derivatives and Their Characteristics

In the current study we performed the synthesis of ethyl 2-amino-1-(furan-2-carboxamido)-5-(2-aryl/tert-butyl-2-oxoethylidene)-4-oxo-4,5-dihydro-1H-pyrrole-3-carboxylates (shown in Figure 1 as 2a-j). These compounds were obtained by recycling furanones 1 under the action of cyanoacetic ester in the presence of a catalyst-triethylamine.

The chemical compounds indicated above were synthesized according to the standard protocols. Briefly, a solution of 0.002 mol of the corresponding 3-(4-methylbenzoyl) hydrazone 2,3-dihydro-2,3-furandione 1 in 20 mL of anhydrous dioxane was added to an equivalent amount of cyanoacetic acid ethyl ester and triethylamine. The resulting mixture was heated for 30–40 min and further cooled to 0 °C. The precipitate was filtered and recrystallized from ethanol.

The resulting compounds 2a-j are yellow-colored crystalline substances, soluble in dimethyl sulfoxide, dimethylformamide, poorly soluble in ethanol, and insoluble in benzene, water, and hexane. The infrared (IR) spectra of synthesized compounds 2a-j are characterized by the presence of absorption bands of stretching vibrations of NH groups in the region of 3109–3414 cm^−1^, carbonyl groups in the region of 1692–1712 and 1646–1676 cm^−1^, as well as stretching vibrations of the C = C double bond in the region of 1590–1628 cm^−1^. In the 1H nuclear magnetic resonance (NMR) spectra, in addition to the signals of substituents in the aromatic ring and the ester group, singlets of protons of the NH group of the amide fragment at 10.47–10.90 ppm, signals of protons of the amino group at 8.85–9.20 ppm and 8.27–8.48 ppm are recorded, respectively; the vinyl proton singlet was recorded at 6.32–6.64 ppm.

Infrared spectra were taken on an IR-Fourier spectrophotometer-spectrometer, model FSM-1201 (SPb Instruments, Saint-Petersburg, Russia) in disks with potassium bromide; 1H NMR spectra were taken on Bruker Avance III apparatus (Bruker Avance, Bremen, Germany) in DMSO-d6, operating frequencies 400 MHz (1H) and 100 MHz (13C), the internal standard is the residual signal from the deuterium solvent. Elemental analysis was performed on a Leco CHNS-932 (Leco Corporation, St. Joseph, MI, USA). The chemical purity of the compounds and the progress of the reactions were monitored by TLC on Sorbfil PTSKh P-A-UF-254 plates (Sorbpolymer, Krasnodar, Russia) in the ether-benzene-acetone solution (ratio 10:9:1); detection was carried out with iodine vapor. Melting points were determined on an SMP40 apparatus (Bibby Scientific Ltd., Stone, UK). 2D structure formulas and spectral characteristics of the synthesized compounds are shown in Appendix A, respectively.

### 3.2. EAPCs Reduce the Viability of the Epithelial Cancer Cell Lines In Vitro

To examine whether EAPCs are active against epithelial cancer cell lines, we performed the MTS-based survival assay by using HCC1806 MDA-MB-231 breast cancer and H1299 non-small cell lung cancer (NSCLC) cancer cells. Each cancer cell line indicated above was treated with various concentrations of EAPCs (0.01–100 μM) for 48–72 h. PTX, Vin and Col were also included as the positive controls and exhibited cytotoxic activities in nanomolar concentrations for all cancer cell lines indicated above. The IC50 values for EAPCs are shown in Table 1, illustrating that EAPC-67, -70, and -71 exhibited the most potent cytotoxic activities for all cancer cell lines included in the present study.

### 3.3. EAPC-67 and -70 Arrest Cancer Cells in M-Phase and Inhibit Tubulin Polymerization

Given that EAPC-67, -70, and -71-treated cancer cells acquired a round-shape morphology (data is not shown), we proposed that the growth-inhibitory activities of these compounds were due to the ability to induce mitotic arrest and interfere with the microtubule dynamic state. To examine this possibility, we performed the western blotting analysis to assess the expression of the proteins specifically accumulated in the M-phase of the cell cycle. Indeed, the expression of pH3 Ser10 and pNuMA Ser395 (the well-known mitotic markers) was significantly increased in EAPC-treated cells, as shown in Figure 2. Importantly, in H1299 NSCLC cells of all three types of EAPCs exhibited much more potent activity to induce cell cycle arrest in M-phase when compared to the well-known TBAs, vinblastine, and colchicine.

To further reveal whether the accumulation of EAPC-treated cancer cells was due to their ability to interfere with the microtubule dynamic state, we performed the tubulin polymerization assay. PTX- and Vin-treated tubulin samples were used as positive and negative controls, respectively. We found that EAPC-67 and -70 effectively inhibited tubulin polymerization, as shown in Figure 3. As expected, PTX exhibited high polymerization activity with tubulin, whereas Vin effectively inhibited tubulin polymerization.

### 3.4. EAPCs Induce Apoptosis of Breast and Lung Cancer Cells

Next, we examined the pro-apoptotic activities of EAPC-67 and -70- in epithelial cancer cell lines. For this purpose, we initially assessed the expression of apoptotic markers (cleaved forms of caspase-3 and PARP) by western blotting. Indeed, we found a significant increase in the expression of apoptotic markers in breast and lung cell lines after EAPC treatment (Figure 4).

Similar to WB data, FACs analysis revealed a significant increase of apoptotic (i.e., Annexin V-positive) cells after EAPC treatment (Figure 5).

Given that EAPC-67 was found to be most active against both of the breast cancer cell lines and also exhibited potent anti-proliferative and cytotoxic activities against the NSCLC cell line (Table 1 and Figure 4 and Figure 5), we further performed computational-based analysis to determine the molecular mechanism of action of this compound.

### 3.5. Molecular Modeling Studies

The molecular modeling study was composed of four parts: (1) Blind docking to determine the most possible binding site in the tubulin for EAPC-67; (2) Binding metadynamics to select the most stable conformation for EAPC-67; (3) Unbiased molecular dynamics of the most stable pose of EAPC-67 within 100 nanoseconds; and (4) Analysis of the structural changes in tubulin upon EAPC-67 binding.

First, we generated EAPC-67 tautomers by using the LigPrep module and further used them for the blind docking to assess their binding with the well-known tubulin binding sites, including colchicine, and the vinca alkaloid, maytansine, and the pironetin binding site. The binding site with the lowest GlideXP score was considered the most possible for ligand binding. Based on the GlideXP score values, we found the CBS as the most energetically favorable for EAPC-67. It is well-known that the protein molecules are not rigid and might undergo conformational changes, in particular after binding with the ligand, thereby suggesting the obvious limitations associated with protein molecule rigidity. Therefore, to more accurately predict the ligand pose, we performed induced-fit docking of two EAPC-67 tautomers with the previously selected CBS.

For each of the EAPC-67 tautomers, 20 ligand poses were generated by the induced docking algorithm. According to the results of induced fit docking, the pose with the maximum IDF score was chosen for each of the two tautomers. To select an EAPC-67 tautomer that is more likely to bind to the tubulin molecule, we performed binding pose metadynamics (BPMD), which allows us to assess the ligand’s stability in solution. Ligand poses that are unstable under the bias of the metadynamics simulation are expected to be infrequently occupied in the energy landscape, thus making minimal contributions to the binding affinity. The results of BPMD were evaluated by the CompScore value, where CompScore is the linear combination of PoseScore and PersScore and is equal to PoseScore − 5 × PersScore. For each of the tautomers, the CompScore was 0.158 and 1.436, respectively (Figure 6). Lower values of the CompScore indicate that the protein-ligand complexes exhibit the highest stability.

Next, we subjected the EAPC-67 tautomer conformation with maximum CompScore and PoseScore to 100 nanosecond unbiased molecular dynamics under periodic conditions. Molecular dynamics was used for the refinement of an IFD Pose and to determine the key interactions between ligand atoms and the amino acids of the CBS. Also, MD allowed us to evaluate the ensemble of available conformations for the complex and generate the “best” single structure approximation (representative conformation) for the complex EAPC-67-tubulin by performing the clusterization assay. An analysis of the structure of the representative conformation provided a better understanding of the mechanism of inhibition of tubulin polymerization induced by the synthesized compounds. The trajectory analysis showed the low RMSD values of the ligands and proteins (Figure 7), which indicates the absence of large fluctuations during the entire time of molecular dynamics.

Over the 100 nanoseconds of molecular dynamics, EAPC-67 predominantly formed the non-covalent contacts with the following amino acids B: Leu 248, B: Asn 249, B: Ala 250, B: Asp 251 of the T7 loop, with B: Lys 254, B: Leu 255 of the H8 helix and with B: Lys 352 of the S9 sheet. Therefore, we found that the ligand interacted predominantly with the amino acids of the secondary elements of the T7 loop, H8 helix, and S9 sheet of CBS. The types of non-covalent interactions with these amino acids, including the formation of hydrogen, hydrophobic, ionic interactions, and the formation of water bridges, as well as the contribution of each interaction type to the overall interaction with a particular amino acid, are shown in the Figure 8A and Figure 9. The time-dependent total number of contacts between the ligand and protein and the number of contacts between the ligand and specific amino acids are shown in Figure 8B.

To select a representative conformation of the ligand-receptor complex, we performed clustering of the molecular dynamics snapshot by the K-means method based on PCA analysis. First, a PCA analysis of the EAPC-67-tubulin complex trajectory was performed, as shown in Figure 10A. This plot demonstrates the distribution of the EAPC-67-tubulin complex conformations in the PC 1–2 subspace, where the blue-white-red color scale reflects the time of the simulation. The contribution of each principal component to the overall variability of atomic positions during molecular dynamics is shown in Figure 10B. The cumulative contribution of the first three PCs cover 15.4%, 26.5%, and 34.5% of the total variance, respectively. Since the number of clusters is critical for the K-means method, we used the Silhouette method to determine their optimal numbers. Based on the Silhouette graph, the optimal number of clusters was determined as 3. Then, clustering of the EAPC-67-tubulin complex conformations by the K-means method was carried out in the PC 1–3 subspace, followed by the projection of the cluster members onto the 2D PC1-2 graph (Figure 10C). The cluster members were further projected onto the RMSD plot to visualize the relationship between conformations that belong to each particular cluster and time. Eventually, we found three large clusters of EAPC-67-tubulin complex conformations that were clearly separated in time with medoids (Snapshots 834, 1556, and 1987) corresponding to each cluster (Figure 10D).

To determine the representative conformation of three medoids, we utilized the MM-GBSA method to evaluate the protein-ligand complex exhibiting the minimal value of ΔGbind prime energy.

As shown in Table 2, the conformation of the EAPC-67-tubulin complex represented as a medoid of cluster 2 exhibited the lowest ΔGbind binding energy and ligand strain energy. The lowest value of ΔGbind indicates that the medoid of cluster 2 is the most probable conformation when compared to the other medoids. The lowest ligand strain energy values also illustrate that minimal energy is required for the adaptation of EAPC-67 to the receptor-bound conformation.

After selecting a representative conformation, we aligned the CBS site to the straight conformation of tubulin obtained from the PDB database (PDB ID: 1JFF). Structural differences were most prominent in the β-chain of tubulin, showing that upon ligand binding the rotation of the T7 loop is necessary. Otherwise, critical steric clashes are formed between the atoms of the ligand and the amino acids of the T7 loop, thereby making the interaction of the protein and the ligand impossible. We also found the displacements of other secondary elements, (e.g., translation of the H7 helix, upward movement, and rotation of S8 and S9 β-sheet) required for the enlargement of free space (Figure 11). Such structural changes increase the available space for the ligand but completely block the transition of the tubulin dimer from the curve to the straight conformation. As a consequence, an effect of the EAPC-67 was due to its ability to interfere with the concerted movements of these secondary structure elements required for tubulin to adopt its straight, microtubular conformation and, therefore, to assemble in the microtubules.

## 4. Discussion

Pyrrole-based compounds are known as one of the attractive scaffolds in medicinal chemistry [39]. Indeed, the privileged structures represent the molecular scaffolds with versatile binding properties, such that a single scaffold can provide potent and selective ligands for a range of different biological targets through modification of the functional groups [40,41]. Therefore, the use of pyrrole–based compounds is considered an attractive strategy to obtain potential drugs due to its ability to modulate several pharmacokinetics parameters, such as solubility, lipophilicity, polarity, and pharmacodynamics, including hydrogen-bonding capacity and the ability to form the complexes with coordinating metals [42,43]. This explains the existence of a broad spectrum of pyrrole-based drugs, including anti-fungal agents, anti-microbial agents, anti-inflammatory agents, HMG-CoA reductase inhibitors, antidepressants, antihypertensive agents, antimalarial agents, anticancer agents, anti-HIV-1 agents, etc. [44,45,46]. To our knowledge, pyrrole and pyrrole-fused heterocycles are suitable to develop novel and effective scaffolds for drugs exhibiting anti-cancer activities due to the targeting of the CBS. Indeed, these scaffolds exhibit high capacities for expanding the chemical space of tubulin inhibitors due to the existence of the various ways to synthesize the derivatives with various functional groups, as well as a wide modification of existing candidate molecules [47,48,49,50,51,52,53].

In addition to the well-established and informative in vitro assays to examine the biological activities of the newly synthesized compounds, computer-aided drug discovery (CADD) has emerged as a powerful and promising technology for cheaper, faster, and effective drug design. These CAAD approaches based on various molecular modeling methods, including standard docking, induced-fit docking, biased/unbiased molecular dynamics, MM-GBSA, etc. are widely used in the field of anti-cancer research [54,55,56]. All of them were shown as effective tools to develop novel inhibitors targeting the various binding sites of proteins, including the CBS of the tubulin. This is due to the availability of the multiple X-ray structures of tubulin-ligand complexes, which provide important information about the conformational changes in the tubulin molecule and interactions between various functional groups of ligands and amino acids of the CBS [57,58,59,60]. The use of the in silico approach significantly reduces the cost to develop novel drugs and allows the negative selection of non-effective molecules at the early stage of drug discovery. Moreover, SAR analysis of various functional chemical groups allows us to find out the novel patterns of interactions between ligand atoms and specific protein amino acids. In addition, such interaction analysis is effective for the prediction of the tubulin depolymerization rate, cytotoxicity against cancer cells, etc. For example, Lorenzo Pallante et al. modeled the β-III tubulin isotype protein structure based on the homology modeling method to discover novel compounds exhibiting high specificity to β-III tubulin isotype [61]. Laura Gallego-Yerga et al. performed a virtual screening of the Zinc database to find potential inhibitors of the colchicine site by using an ensemble docking approach, molecular dynamics, and pharmacophore modeling. As a result, the authors synthesized tetrazole derivatives and further tested their potential activity to interfere with tubulin polymerization in vitro. In good agreement with the design principles, this compound demonstrated potent in vitro activity against tubulin polymerization in a micromolar range and an anti-proliferative effect against human epithelioid carcinoma HeLa cell line in nanomolar concentrations [62]. Based on the 2D-quantitative SAR analysis of quinolines and using the docking method, Mohamed et al. synthesized 29 quinoline derivatives targeting the CBS and exhibiting potent anti-proliferative activities against various human cancer cell lines including colorectal carcinoma (HCT-116), hepatocellular carcinoma (HepG-2), and breast cancer (MCF-7) [63]. In addition, abundant examples illustrating the effectiveness of computer simulation modeling approaches for the design and synthesis of tubulin inhibitors targeting the CBS were shown and reviewed recently [64,65,66,67].

In agreement with current studies, we used a combination of computer modeling methods indicated above and biological activity assays to discover the anti-cancer activities of pyrrole-based compounds targeting the CBS of tubulin. As a result of our CAAD approaches, we report here about the novel pyrrole-based active compounds exhibiting potent cytotoxic activities against distinct cancer cell lines, including HCC1806, MDA-MB-231 triple-negative breast cancer, and H1299 non-small cell lung cancer cells. Among all synthesized compounds, EAPC-67 and EAPC-70 were the most active, demonstrating the highest cytotoxic activities against cancer cells in a dose-dependent manner. Next, by using the polymerization assay, we found that the cytotoxic activities of EAPCs are due to their ability to interfere with the microtubule dynamic state (Figure 3). Molecular modeling data revealed that the EAPCs interact directly with the CBS and induce the conformational changes of the tubulin dimer that are typical to the mode of action of the compounds targeting the CBS [68,69]. In particular, translation of the H7 helix, upward movement, rotation of the S9 β-sheet, and rotation of the T7 loop in the β-chain of tubulin was demonstrated. The transition of these secondary elements is impossible to the position characteristic of the straight tubulin conformation in the assembled microtubule because the EAPCs occupy the physical space between the α- and β-chains. Therefore, EAPCs-bound free tubulin molecules cannot assemble into organized microtubules (Figure 11). Our molecular modeling data was a proper fit with the biological activities observed for EAPCs. Indeed, we found that EAPC-67 and -70 effectively inhibited tubulin polymerization. As an outcome of EAPC-induced failure of assembly of microtubules, we observed a robust cell cycle arrest in the G2/M phase and the accumulation of cells in the M-phase. This was evidenced by the substantial increase of pH3 Ser10 and pNuMA Ser395 expression in cancer cells treated with EAPCs (Figure 2). As a result of the mitotic arrest, EAPCs-treated cells underwent apoptotic cell death, which was evidenced by a substantial increase of the cleaved forms of caspase-3 and PARP in the vast majority of cancer cell lines and also a significant increase of Annexin V-positive cells after EAPC treatment (Figure 4 and Figure 5, respectively). Collectively, new synthesized EAPCs compounds, in particular EAPC-67 and EAPC-70, can be used as a potential scaffold for the development of new and more promising MTAs targeting the colchicine-binding site.

## Figures and Tables

**Figure 1 molecules-27-02873-f001:**
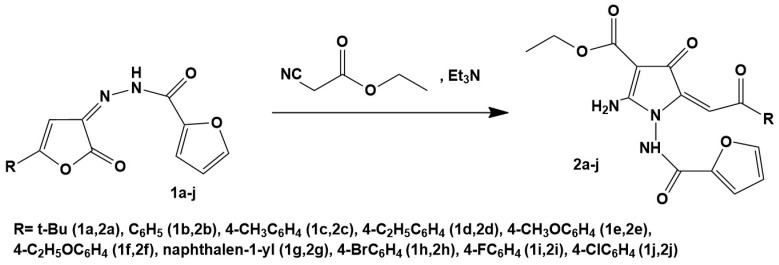
The common synthetic pathway of the derivatives of 2-amino pyrroles.

**Figure 2 molecules-27-02873-f002:**
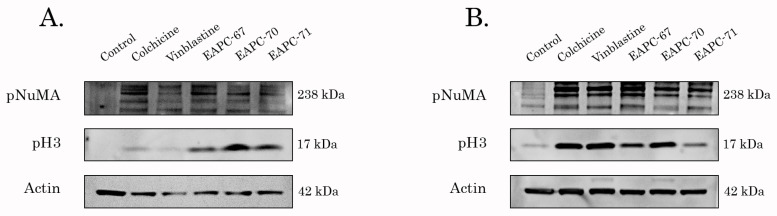
Immunoblot analysis for the expression of M-phase specific proteins (e.g., phospho-NuMA Ser395 and phospho-H3 Ser10) in H1299 lung cancer (**A**) and HCC1806 breast cancer (**B**) treated with DMSO (negative control), EAPC-67, EAPC-70, EAPC-71 (both 10 μM), Colchicine (0.05 μM) and Vinblastine (0.01 μM) for 24 h. Actin stain was used as a loading control.

**Figure 3 molecules-27-02873-f003:**
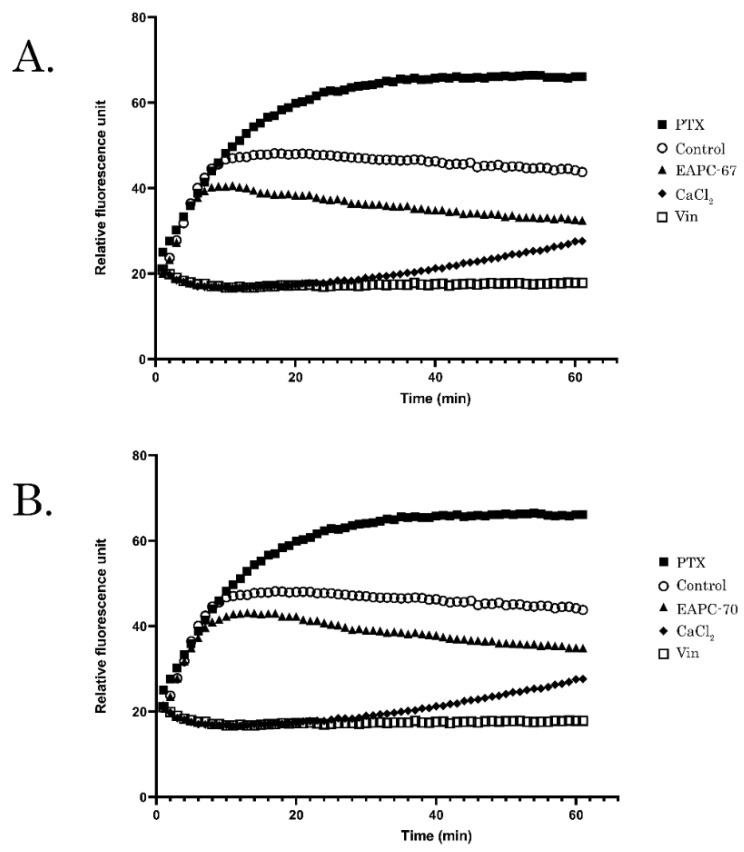
Dynamics of tubulin polymerization in samples treated with EAPC-67 (**A**) and EAPC-70 (**B**). Tubulin was also incubated with DMSO (control), Paclitaxel (PTX), Vinblastine (Vin), and CaCl2 at 37 °C, and absorbance was assessed every min for 1 h. A shift of the curve to the upper left of the control (DMSO) represents an increase in polymerized microtubules. A shift to the bottom right of the graph reflects the decrease in the rate of tubulin polymerization.

**Figure 4 molecules-27-02873-f004:**
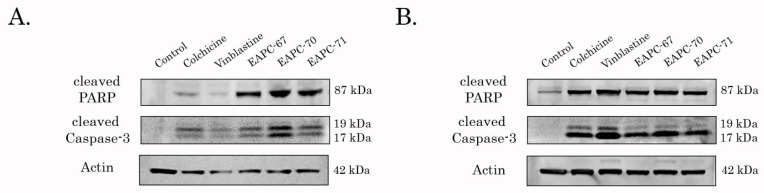
Immunoblot analysis for apoptosis markers (e.g., cleaved forms of PARP and caspase-3) in H1299 lung cancer (**A**) and HCC1806 breast cancer (**B**) after treatment with DMSO (negative control), EAPC-67, EAPC-70, EAPC-71 (10 μM), colchicine (0.05 μM) and vinblastine (0.01 μM) for 24 h. Actin stain is used as a loading control.

**Figure 5 molecules-27-02873-f005:**
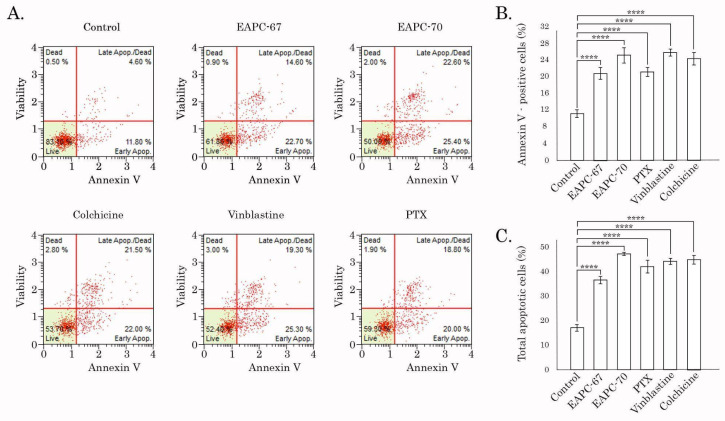
FACs analysis for apoptotic markers in HCC 1806 breast cancer cells treated with DMSO (control), paclitaxel, vinblastine, and colchicine (positive control), EAPC-67, and EAPC-70 for 24 h. (**A**) Representative dot plots are shown. (**B**) Quantitative analysis of the early-apoptotic cells after the treatment as indicated above. (**C**) Quantitative analysis of the total apoptotic cells after the treatment as indicated above. **** *p* < 0.0001.

**Figure 6 molecules-27-02873-f006:**
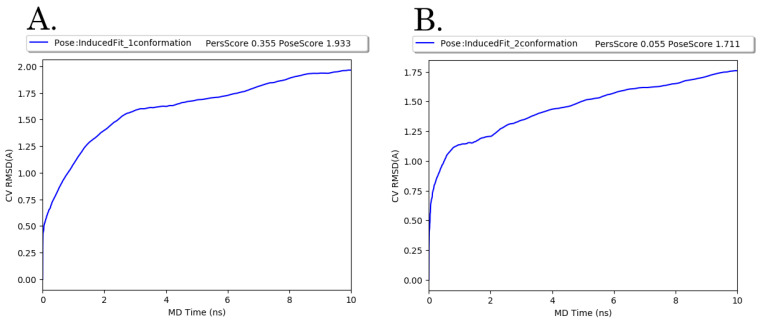
Average RMSD of two EAPC-67 (**A**,**B**) tautomers conformation over 10 × 10 ns metadynamics runs with calculated values of PoseScore and PersScore.

**Figure 7 molecules-27-02873-f007:**
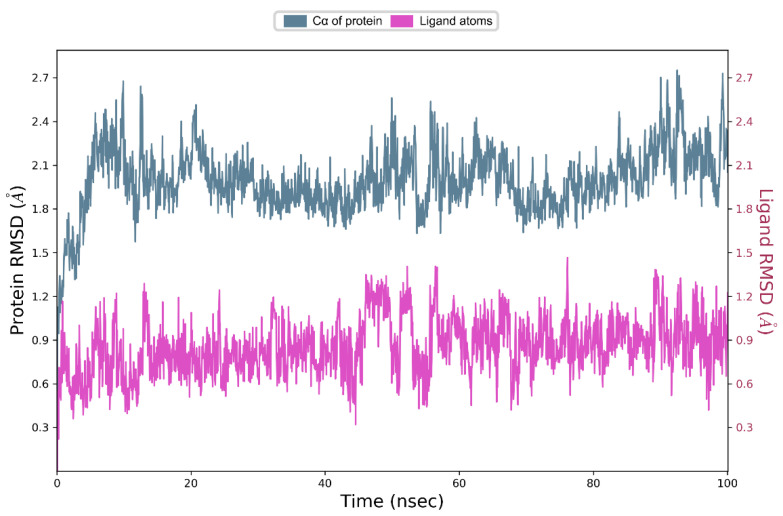
Change in RMSD (root means square deviation) for ligand (pink color) and for Cα atoms of tubulin dimer (blue color) during 100 ns of unbiased molecular dynamic simulation.

**Figure 8 molecules-27-02873-f008:**
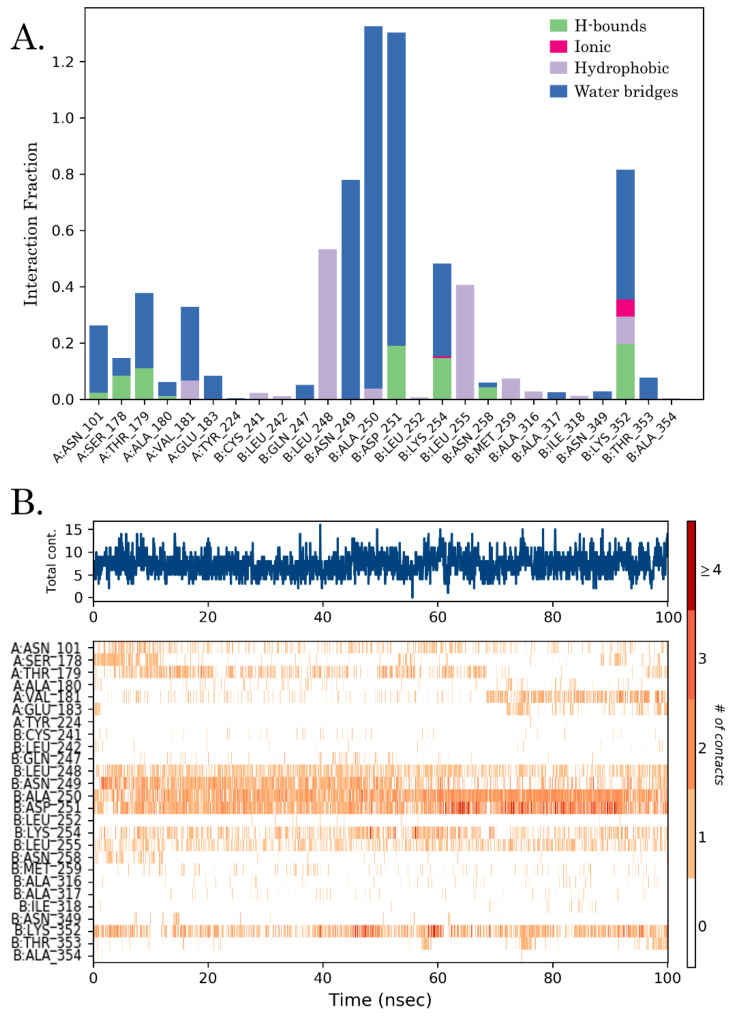
(**A**) Protein-ligand fraction interaction diagram between EAPC-67 and tubulin. H-bonds, ionic, hydrophobic interactions, and water bridges are shown in green, red, purple, and blue, respectively. (**B**) Contact points of EAPC-67 and amino acids of colchicine binding site during the whole simulation trajectory. The top panel shows the total number of protein-ligand contacts over 100 ns.

**Figure 9 molecules-27-02873-f009:**
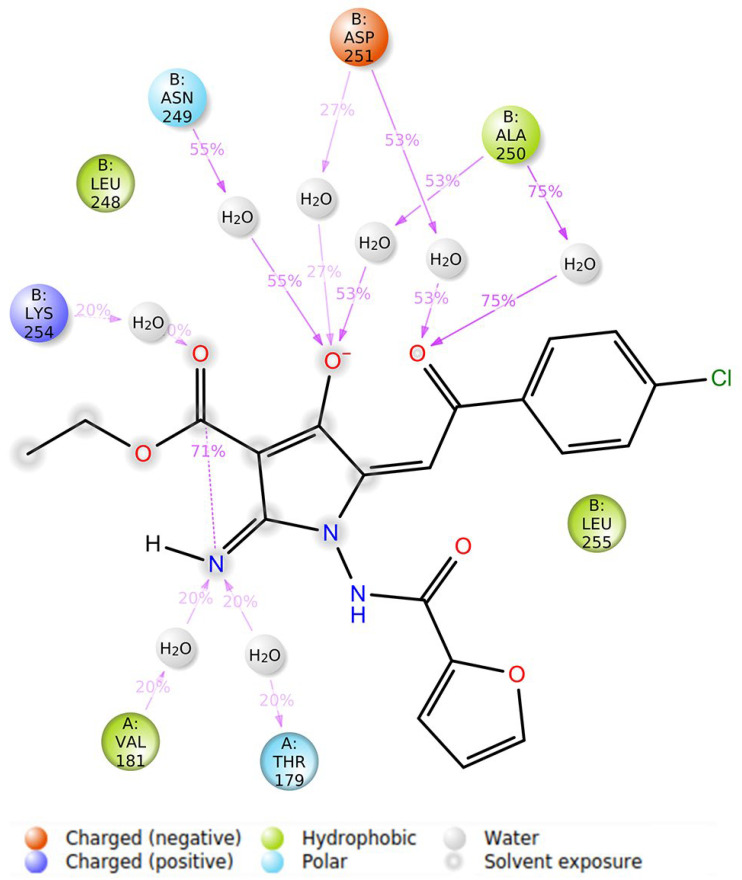
2D plot of ligand-protein fraction interaction diagram between EAPC-67 and tubulin over 100 nanoseconds MD with labeled key amino acids. Each type of non-covalent interaction is highlighted in the corresponding color.

**Figure 10 molecules-27-02873-f010:**
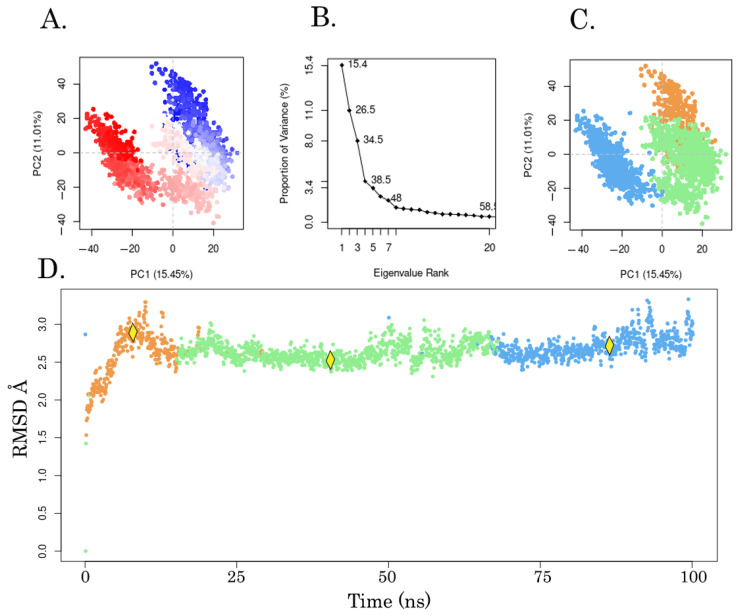
Principal component analysis (PCA) of EAPC-67-tubulin complex trajectory. (**A**) Plotting PC1 component (*x*-axis) versus PC2 component (*y*-axis). Each dot denotes the one conformation of the protein. The spread of blue and red color dots describes the degree of conformational changes in the simulation, where the color scale from blue to white to red is equivalent to the simulation time. The blue indicates the initial timestep, white is intermediate and the final timestep is represented by the red color. (**B**) The proportion of variance per each principal component. (**C**) Clustering of the EAPC-67-tubulin complex conformations in the PC1-PC3 subspace based on the K-means method and projection of cluster members onto the PC1-PC2 plot diagram. A member of each of the three clusters is indicated by a separate color. (**D**) Conformational clusters of the EAPC-67-tubulin complex are displayed on the RMSD plot. Each dot represents a complex conformer, and the yellow markers indicate the representative conformations (medoids–snapshots 834, 1556, and 1987) of each cluster.

**Figure 11 molecules-27-02873-f011:**
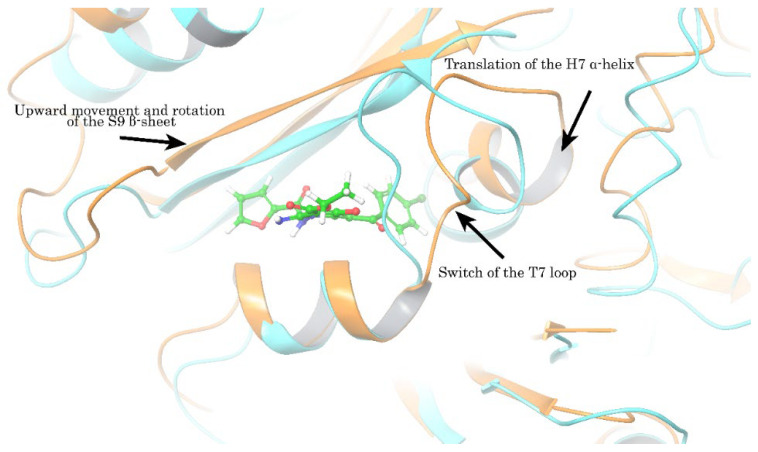
The structural difference between straight (PDB ID: 1JFF) and representative conformations of β-chain of tubulin was obtained from MD. The straight conformation (light brown color) of the β-chain of tubulin is aligned with the representative conformation (blue color). Arrows indicate the main secondary elements undergoing significant structural changes during the transition from one conformation to another. The straight confirmation is not suitable for interaction with the EAPC-67, since steric clusters are formed with the amino acids of the protein secondary elements (S8, S9, H7, T7) surrounding the colchicine-binding pocket.

**Table 1 molecules-27-02873-t001:** IC_50_ values (in the micromolar range) for EAPCs in HCC1806, MDA-MB-231 triple-negative breast cancer, and H1299 non-small cell lung cancer cell lines.

Number of EAPC	Cell Lines		
	MDA-MB 231	H1299	HCC 1806
64	21.5 ± 2.01	28.7 ± 1.54	23.19 ± 3.72
65	28.2 ± 1.12	17.5 ± 2.31	14.13 ± 2.91
66	40.2 ± 3.42	46.3 ± 4.21	54.2 ± 4.87
67	2.9 ± 0.21	6.7 ± 0.53	9.4 ± 0.64
68	>100	>100	56.16 ± 3.2
69	26.1 ± 2.56	21.5 ± 1.5	29.50 ± 1.43
70	6.7 ± 0.52	2.2 ± 0.25	25.35 ± 2.12
71	10.24 ± 1.36	2.5 ± 0.42	29.73 ± 1.91
72	85.1 ± 1.25	79.6 ± 2.5	68.63 ± 4.17
73	39.3 ± 2.97	>100	>100

**Table 2 molecules-27-02873-t002:** ΔGbind energy and ligand strain energy values for each medoid of three clusters.

Medoids of Cluster	MM-GBSA ΔG bind (kcal/mol)	Ligand Strain Energy (kcal/mol)
Snapshot 834	−24.2	11.4
Snapshot 1556	−31.1	9.3
Snapshot 1987	−25.6	10.9

## Data Availability

Not applicable.

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
