# Peer review of "Computational-Based Discovery of the Anti-Cancer Activities of Pyrrole-Based Compounds Targeting the Colchicine-Binding Site of Tubulin"

_molecules, 2022, doi:10.3390/molecules27092873_

Round 1

Reviewer 1 Report

Dear S. Boichuk et al,

The research article entitled "Computational-based discovery of the anti-cancer activities of pyrrole-based compounds targeting the colchicine-binding site of tubulin" was submitted to molecules. The article presented very well, but there are a few things that the authors have to discuss extensively based on the computational findings.

  1. Fig 7, The ligand and protein RMSD are stable, but how is this related to the binding pocket?
  2. Fig 9 should be explained extensively
  3. The 2D plot of the MDS plot is missing
  4. MM-GBSA impact, result not discussed
  5. What is the role of Metadynamics in this study?
  6. Colchicine binding site is well studied in previous research, moreover, co-crystal of colchicine with tubulin is present on the PDB website. So what is the rationale behind this study?

Author Response

We appreciate a reviewer for the comments and suggestions regarding our manuscript. Below are our specific responses to these comments (shown in quotes and italics). All the changes in the revised version of the manuscript are highlighted with yellow.

Dear S. Boichuk et al,

The research article entitled "Computational-based discovery of the anti-cancer activities of pyrrole-based compounds targeting the colchicine-binding site of tubulin" was submitted to molecules. The article presented very well, but there are a few things that the authors have to discuss extensively based on the computational findings.

  1. “Fig 7, The ligand and protein RMSD are stable, but how is this related to the binding pocket?”. We agree with a reviewer that RMSD data illustrating the EAPC-67-tubulin complex during 100 nanoseconds does not related directly with the conformational changes in the binding site. However, RMSD of tubulin dimer in presence of EAPC-67 demonstrates stability during the whole trajectory without any spikes, which allows us to make a conclusion about the stability of EAPC-67-tubulin complex.
  2. “Fig 9 should be explained extensively”. We appreciate a reviewer for this suggestion and performed these changes in the manuscript as recommended.
  3. “The 2D plot of the MDS plot is missing”. We greatly appreciate a reviewer for this comment and introduced 2D plot into the manuscript as was recommended.
  4. “MM-GBSA impact, result not discussed”. We introduced these changes into the revised version of the manuscript as was recommended by a reviewer. Thank you!
  5. “What is the role of Metadynamics in this study?” We used the Binding pose metadynamics algorithm to choose the most stable conformation between two EAPC-67 tautomeres, that were previously generated by using the induced fit docking. Binding pose metadynamics is a well-known method to assess the stability of the ligand pose. This method allows to choose the tautomers conformations which are very close to each other in terms of scoring function value obtained by induced fit docking. Thus, the use of Binding pose metadynamics allowed us to choose the most stable tautomer and to perform the detailed analysis of ligand-protein interactions for this tautomer during 100 nanoseconds. This, in turn, allowed us generate our “best” single structure approximation for the complex. A similar algorithm is used in IFD-MD procedure in Schrodinger Maestro software [1].

1.Edward B. Miller, Robert B. Murphy, Daniel Sindhikara, Kenneth W. Borrelli, Matthew J. Grisewood, Fabio Ranalli, Steven L. Dixon, Steven Jerome, Nicholas A. Boyles, Tyler Day, Phani Ghanakota, Sayan Mondal, Salma B. Rafi, Dawn M. Troast, Robert Abel, and Richard A. FriesnerJournal of Chemical Theory and Computation 2021 17 (4), 2630-2639 DOI: 10.1021/acs.jctc.1c00136

  1. “Colchicine binding site is well studied in previous research, moreover, co-crystal of colchicine with tubulin is present on the PDB website. So what is the rationale behind this study?” We agree with a reviewer for this comment. Indeed, the structure of the colchicine-binding site is a well-known through multiple approaches including X-ray based crystallography, cryoelectron microscopy, etc used in structural biology. The aim of our study was not to study the structural features of the colchicine binding site. We are illustrating here that pyrrole-based tubulin inhibitors are effectively binding to the colchicine-binding site and these changes are very similar to the colchicine-induced changes in the tubulin dimer.

Reviewer 2 Report

The manuscript titled “Computational-based discovery of the anti-cancer activities of pyrrole-based compounds targeting the colchicine-binding site of tubulin” needs revision before it can be accepted.Major concerns are highlighted, that must be rectified.

  1. potent anti-cancer activities against the breast and lung cancer cell lines in vitro. Italicise
  2. The microtubules are known as important regulators of many aspects of cellular 43 functions, including cell proliferation and migration, vesicle trafficking during endocyto- 44 sis, chromosomal segregation during mitosis, etc. Cite a relevant citation. Such sentences can’t be put up as such.
  3. Finally, based on the metadynamics data, we performed 100 nanosecond. The dynamics simulation time should be carried out for prolonged time to understand the nature of the complexes and give a proper conclusion.
  4. Authors have to describe the aim of the study clearly. It seems to be missing and the authors are advised to add a paragraph for the same in the Introduction section.
  5. I recommend the authors crosscheck the manuscript for the abbreviations used in the first place. The use of abbreviations is erratic, and it is of no benefit to using abbreviations if they are not used consistently throughout the manuscript..
  6. Did the authors carry out energy minimization?
  7. The scientific problem is presented in an interesting way. However, there are a few language mistakes in the text. Language editing is required to improve the quality of the manuscript.
  8. What is the significance of MD simulations in this study? A brief description must be provided.
  9. A bunch of papers on the similar work are reported. How Author justified the novelty and significance of current findings?
  10. Discussion should be improved in light of authors findings. I would suggest being polite and avoiding writing strong sentences.
  11. The molecular mechanics/generalized Born surface area (MM-GBSA) 182 method was used as a post-molecular dynamic validation tool. Did the authors tried using LIE?
  12. What was the basis to use lung cancer cell lines?
  13. The authors should comment about how many poses were obtained by the docking and the basis for choosing the bet pose
  14. Other studies have also reported potent anti-cancer compounds using computational based approaches. These can be incorporated at relevant places.

https://doi.org/10.1016/j.molliq.2022.118581

https://doi.org/10.1038/s41598-020-65648-z

https://doi.org/10.3390/molecules25040823

Author Response

We appreciate a reviewer for the comments and suggestions regarding our manuscript. Below are our specific responses to these comments (shown in quotes and italics). All the changes in the revised version of the manuscript are highlighted with yellow.

The manuscript titled “Computational-based discovery of the anti-cancer activities of pyrrole-based compounds targeting the colchicine-binding site of tubulin” needs revision before it can be accepted. Major concerns are highlighted, that must be rectified”.

  1. potent anti-cancer activities against the breast and lung cancer cell lines in vitro. Italicise”. We consider these changes are not necessary and asking a reviewer to leave them non-changed.
  2. The microtubules are known as important regulators of many aspects of cellular functions, including cell proliferation and migration, vesicle trafficking during endocytosis, chromosomal segregation during mitosis, etc. Cite a relevant citation. Such sentences can’t be put up as such”. We included the citations illustrating the physiological role of the microtubules in revised manuscript according this recommendation.
  3. Finally, based on the metadynamics data, we performed 100 nanosecond. The dynamics simulation time should be carried out for prolonged time to understand the nature of the complexes and give a proper conclusion”. We disagree with this suggestion of the reviewer. Indeed, molecular dynamics performed for 100 nanoseconds is a common and well-established approach used for dynamics simulation of the ligand-protein complexes. This time-frame (100 nanoseconds) is very commonly used and evidenced by the multiple manuscripts in peer-reviewed journals [1-3], including the manuscripts suggested to be cited by a reviewer [4, 5].

1.Londhe AM, Gadhe CG, Lim SM, Pae AN. Investigation of Molecular Details of Keap1-Nrf2 Inhibitors Using Molecular Dynamics and Umbrella Sampling Techniques. Molecules. 2019 Nov 12;24(22):4085. doi: 10.3390/molecules24224085. PMID: 31726716; PMCID: PMC6891428.

  1. Li L, Wang Q, Zhang Y, Niu Y, Yao X, Liu H. The molecular mechanism of bisphenol A (BPA) as an endocrine disruptor by interacting with nuclear receptors: insights from molecular dynamics (MD) simulations. PLoS One. 2015;10(3):e0120330. Published 2015 Mar 23. doi:10.1371/journal.pone.0120330
  2. Tumskiy RS, Tumskaia AV. Multistep rational molecular design and combined docking for discovery of novel classes of inhibitors of SARS-CoV-2 main protease 3CLpro. Chem Phys Lett. 2021 Oct;780:138894. doi: 10.1016/j.cplett.2021.138894. Epub 2021 Jul 14. PMID: 34276059; PMCID: PMC8277558.
  3. Mohammad T, Siddiqui S, Shamsi A, Alajmi MF, Hussain A, Islam A, Ahmad F, Hassan MI. Virtual Screening Approach to Identify High-Affinity Inhibitors of Serum and Glucocorticoid-Regulated Kinase 1 among Bioactive Natural Products: Combined Molecular Docking and Simulation Studies. Molecules. 2020 Feb 13;25(4):823. doi: 10.3390/molecules25040823. PMID: 32070031; PMCID: PMC7070812.
  4. Anwar S, DasGupta D, Shafie A, et al. Implications of tempol in pyruvate dehydrogenase kinase 3 targeted anticancer therapeutics: Computational, spectroscopic, and calorimetric studies. Journal of Molecular Liquids. 2022;350:118581. doi:10.1016/j.molliq.2022.118581

  1. Authors have to describe the aim of the study clearly. It seems to be missing and the authors are advised to add a paragraph for the same in the Introduction section”. We expanded the Introduction section to clarify the aim of the study, as recommended by a reviewer.
  2. I recommend the authors crosscheck the manuscript for the abbreviations used in the first place. The use of abbreviations is erratic, and it is of no benefit to using abbreviations if they are not used consistently throughout the manuscript.” We looked over the manuscript very carefully and performed these changes in revised version of the manuscript.
  3. “Did the authors carry out energy minimization?”. Energy minimization was performed for the ligand and for the protein by using the OPLS4 force field. The energy minimization step is integrated into the LigPrep and Protein Preparation Wizard modules in Schrodinger Maestro software, which was used for ligand and receptor preparation for molecular docking, metadynamics, etc.
  4. The scientific problem is presented in an interesting way. However, there are a few language mistakes in the text. Language editing is required to improve the quality of the manuscript”. We appreciate a reviewer for this comment and looked over the manuscript very carefully and performed the language editing.
  5. What is the significance of MD simulations in this study? A brief description must be provided”. Molecular dynamics was used for the refinement of an IFD Pose and to generate our “best” single structure approximation for the complex EAPC-67-tubulin. The description of molecular dynamics used to study the interactions between EAPC-67 and tubulin dimer to generate the representative conformation was introduced in the Results section, as recommended by a reviewer.
  6. A bunch of papers on the similar work are reported. How Author justified the novelty and significance of current findings?”. We agree with a reviewer – multiple manuscripts illustrating the high potency of newly-synthesized chemical compounds targeting the microtubules were published recently. Our work highlights the novel pyrrole-based structures (synthesized in our lab) exhibiting high potency to bind with the colchicine-binding site of tubulin and thereby induce cell cycle arrest and apoptosis in various cancer cell lines in vitro. This in turn allows us to propose that these compounds can be used as a prospective scaffold to generate a novel class of anti-cancer agents active against cancer cells overcoming
  7. Discussion should be improved in light of authors findings. I would suggest being polite and avoiding writing strong sentences”. We expanded a discussion section and performed the changes according the reviewer’s recommendations.
  8. The molecular mechanics/generalized Born surface area (MM-GBSA) 182 method was used as a post-molecular dynamic validation tool. Did the authors tried using LIE?” We agree with a reviewer that the linear interaction energy (LIE) approach might be used as an alternative for MM-GBSA to calculate the free binding energy. However, we did not use LIE approach for 2 reasons. First, recent studies demonstrated the both methods demonstrate the similar predictive values for free binding energy [1]. Second, LIE calculation is performed by using the Gromacs software, whereas we utilized the DESMOND module in Schrodinger Maestro software for our molecular dynamics study. To avoid the loss of data which might happen during the conversion of the formats cms/maestro into gro by using the VMD (Visual Molecular Dynamics) software, we decided to use MM-GBSA method integrated into the Schrodinger Maestro software.
  9. Rifai EA, van Dijk M, Vermeulen NPE, Yanuar A, Geerke DP. A Comparative Linear Interaction Energy and MM/PBSA Study on SIRT1-Ligand Binding Free Energy Calculation. J Chem Inf Model. 2019 Sep 23;59(9):4018-4033. doi: 10.1021/acs.jcim.9b00609. Epub 2019 Sep 11. PMID: 31461271; PMCID: PMC6759767.

  1. What was the basis to use lung cancer cell lines?” Since the epithelial cancer cell lines are sensitive to the tubulin-binding agents (TBAs) we utilized breast and lung cancer cell lines as the most appropriate and suitable cancer cells for this study.
  2. The authors should comment about how many poses were obtained by the docking and the basis for choosing the bet pose”. We appreciate a reviewer for this comment and included the info about the number of poses into the Results section.
  3. Other studies have also reported potent anti-cancer compounds using computational based approaches. These can be incorporated at relevant places”. We agree with a reviewer and included the suggested references into the relevant places.

After making these changes according to reviewer’s recommendation we are now submitting our revised manuscript for your kind consideration of publication in Molecules.

Reviewer 3 Report

The manuscript titled “Computational-based discovery of the anti-cancer activities of pyrrole-based compounds targeting the colchicine-binding site of tubulin” used in vitro and in silico studies to identify potential anti-cancer compounds. Overall, the work is well designed and executed but lacks a detailed explanation in the molecular dynamics simulations. The results need to be explained in molecular detail. Most of the figures are low quality, cropped and they are not clearly presented.

Author Response

We appreciate a reviewer for the comments regarding our manuscript. Below are our specific responses to these comments (shown in quotes and italics).

The manuscript titled “Computational-based discovery of the anti-cancer activities of pyrrole-based compounds targeting the colchicine-binding site of tubulin” used in vitro and in silico studies to identify potential anti-cancer compounds. Overall, the work is well designed and executed but lacks a detailed explanation in the molecular dynamics simulations. The results need to be explained in molecular detail. Most of the figures are low quality, cropped and they are not clearly presented.

We agree with a reviewer and included a detailed explanation in the molecular dynamics simulation in the corresponding sections of the manuscript (Results and Discussion, as well). The results of the studies were explained in molecular detail in Results and Discussion, as was recommended. The quality of the figures was also improved. The changes in the revised version of the manuscript are highlighted with yellow.

After making these changes according to reviewer’s recommendation we are now submitting our revised manuscript for your kind consideration of publication in Molecules.

Reviewer 4 Report

In this study, the author designed and synthesised EAPC detivatives and also prososed the likely binding mode. Cellular study also inidcated similar molecular behaviours as TBAs. This work includes decent amount of work. However, if the author can use the classical in vitro TBA binding assay to show the binding activity of this series of compounds, this result will be more solid.

Author Response

We greatly appreciate a reviewer for the comments and suggestions regarding our manuscript. Below are our specific response to these comments (shown in quotes and italics).

In this study, the author designed and synthesized EAPC derivatives and also proposed the likely binding mode. Cellular study also indicated similar molecular behaviours as TBAs. This work includes decent amount of work. However, if the author can use the classical in vitro TBA binding assay to show the binding activity of this series of compounds, this result will be more solid.

We agree with a reviewer that the use of classical TBA binding assay will be informative and make the data more solid. At the same time, we demonstrate here the functional activities of EAPC derivates and show their ability to interfere deeply with the microtubules dynamic state by inhibiting the tubulin polymerization.  This, in turn, leads to the abnormalities in the cell cycle regulation and further induces the apoptosis in cancer cells. To our opinion, this data (together with the computational-based approaches utilized for this manuscript) allowed us to argue about the tubulin-binding activities of EAPC derivates. These activities will be further examined both by using the in vitro and in vivo assays, including TBA binding assay, as well.

The changes in the revised version of the manuscript are highlighted with yellow. After making these changes according to reviewer’s recommendations we are now submitting our revised manuscript for your kind consideration of publication in Molecules.

Round 2

Reviewer 2 Report

The authors have made the changes as suggested and manuscript can now be accepted for publication.